

# Comparison of the performance of skip lists and splay trees in classification of internet packets

Navid Khezrian[1] and Mahdi Abbasi[2]

[1] Computer Engineering Faculty, Sharif University of Technology, Tehran, Iran
[2] Department of Computer Engineering, Engineering Faculty, Bu-Ali Sina University, Hamedan, Iran

## ABSTRACT

Due to the increasing number of Internet users and the volume of information exchanged by software applications, Internet packet traffic has increased significantly, which has highlighted the need to accelerate the processing required in network systems. Packet classification is one of the solutions implemented in network systems. The most important issue is to use an approach that can classify packets at the speed of the network and show optimum performance in terms of memory usage. In this study, we evaluated the performance in packet classification of two of the most important data structures used in decision trees, i.e. the skip list and splay tree. Our criteria for performance were the time of packet classification, the number of memory accesses, and memory usage of each event. These criteria were tested by the ACL and IPC rules with different numbers of rules as well as by different packet numbers. The results of the evaluation showed that the performance of skip lists is higher than that of splay trees. By increasing the number of classifying rules, both the difference in the speed of packet classification and the superiority of the performance of the skip list over that of the splay tree become more significant. The skip list also maintains its superiority over the splay tree in lower memory usage. The results of the experiments confirm the scalability of this method in comparison to the splay tree method.

# INTRODUCTION

The Internet is the largest packet-switching network. In this network, information is transmitted in the form of packets from the source to the destination. With the increase in the number of users and the volume of information exchanged by applications, Internet packet traffic has increased significantly. For this reason, in order to accelerate the processing required in network systems such as routers, a basic process called packet classification is used (*Baboescu & Varghese, 2001*; *Taylor, 2005*; *Perez et al., 2014*). Classification of network packets refers to the different streams of packets in network systems (*Bontupalli et al., 2018*; *Harada, Tanaka & Mikawa, 2018*; *Inoue et al., 2018*; *Li et al., 2018*; *Bi, Luo & Sun, 2019*). Many network systems use packet routing and guiding policies as well as quick implementation of packet classification policies to carry out traffic management policies (*Li et al., 2013*; *Lin et al., 2016*; *Tessier et al., 2018*). Using these basic processes, packet flow

Corresponding author
Mahdi Abbasi, abbasi@basu.ac.ir

processing has become possible at a very high speed, and the same rules can be applied to all packets belonging to the same traffic stream (*Comer, 2004*). Network applications that require packet classification are of three types, i.e., security operations, traffic management and quality of service (QOS), and policy-based routing. Several studies have analytically or experimentally have benchmarked the different algorithms of packet classification (*Gao, Tan & Gong, 2006*; *Qi et al., 2009*; *Lim, 2010*; *Nagpal et al., 2015*). A commonly accepted categorization of the packet classification algorithms is the one presented by *Taylor (2005)*. According to this categorization, the packet classification fall into four classes which are explained below.

### Exhaustive search

In this type of algorithm, all elements within a list are checked to match the search query argument. The main disadvantage of these algorithms is the linear dependence of time complexity on the number of filters (*Trabelsi & Zeidan, 2011*).

### Decomposition

In decomposition-based algorithms, two processing steps are followed. In the first step, the search is performed individually on the filter set based on each field. In the second step, the results of all searches on the different fields are merged through intersection (*Neji & Bouhoula, 2008*). Therefore, it is obvious that these algorithms have great potential for parallelism. However, the large size of the data structures required in these algorithms makes them inefficient in terms of memory usage.

### Tuple spaces

In this method, the filters are divided by the number of bits specified in the prefixes of the search query, and the search space is thus partitioned into several sub-spaces. During classification, the input packets are carefully matched and checked against the generated tuples using the simple or tree-based search algorithm on the prefix fields of interest (*Kirschenhofer, Martínez & Prodinger, 1995*). When matching a packet with a tuple is successful, only those filters are evaluated that are in the equivalent sets of the tuple with regard to their matching with other fields of the packet. The memory complexity of these algorithms is less than the decomposition-based algorithms (*Srinivasan, Suri & Varghese, 1999*).

### Decision tree

In these algorithms, the set of filters is stored in search trees based on the binary patterns in the prefix fields of the filters. To make a decision tree based on several fields, a tree is created in which the leaves contain a specified filter or a subset of filters that have an intersection in the traversed prefix from the root to the leaves. In these algorithms, the best filter corresponding to the input package is found through the binary contents of the fields in question on the search tree (*Sen, 1991*).

The existing methods have not been able to balance the time and memory consumption. On the other hand, binary trees work well when the elements enter accidentally, but they become inefficient in cases where the operations are sequential. Tree algorithms

use different data structures for searching. Two of the most important data structures of decision trees are the splay tree (*Sleator & Tarjan, 1985*) and the skip list (*Kaufmann, 2007*). The performance of a splay tree depends on the history of accesses to its elements. On the other hand, the performance of a skip list depends on an independent randomization of the height of links that lead to specific elements. Therefore, probabilistic methods are used to analyze the operation of splay trees and skip lists. We refer the reader to references (*Papadakis, 1993*; *Pugh, 1990*; *Sen, 1991*; *Papadakis, 1993*; *Kirschenhofer, Martínez & Prodinger, 1995*) for probabilistic analysis of the complexity of these algorithms.

In this paper, we intend to evaluate and compare the performance of packet-classifying tree algorithms using these two different data structures. For this purpose, we will use the criteria of time complexity and memory complexity. Time complexity depends on the number of algorithmic references to memory to classify each packet and memory complexity depends on the amount of memory used by the data structure of the algorithm.

The structure of the paper is organized as follows. First, we review the history of packet-classifying tree algorithms and related previous works for evaluating their performance. The third section describes the general structure of the tree algorithms based on skip lists and splay trees along with their implementation. In the fourth section, after introducing the tools used to produce filters and packets, the evaluation criteria are presented and the results of the evaluation of the performance of the two approaches are compared. The final section draws conclusions and indicates directions for further research.

## Background

The main aim of the paper is to compare the performance of the skip list and splay tree data structures when adapted to multidimensional search on the rule set of a packet classifier. The nature of the search, insert and update of such data structures lets tree-based packet classifiers to reduce the number of required memory during search and hence reduce the complexity of classification.

A review of recent research suggests that no study so far has conducted to make an in-depth comparison of the performance of packet-classifying tree algorithms operating with skip lists and splay trees. Previous works simply aimed at optimizing these algorithms without comparing their performance.

*Pan et al. (2016)* used the skip list in 2016 to improve the time performance of information retrieval algorithms in local lists. In their design, given that a packet might share a prefix with previous packages, search in the skip list starts from the closest node previously obtained from this prefix. Therefore, a significant amount of time is saved. Extensive evaluations show that their design can triple the speed of the original design on a 32-bit machine.

In 2015, *Trabelsi et al. (2015)* proposed a multi-stage and dynamic packet filtering mechanism to enhance the performance of the firewall. Their proposed mechanism is implemented by splay tree filters and uses traffic features to minimize packet filtering time. It can decide whether or not dynamic updates of the splay tree filters are needed to filter the next network traffic window and predict the best customized pattern for the tree. In this method of input packet filtering, the initial acceptance of the packet is done using

splay tree data structure, which is dynamically updated according to the traffic streams of the network. As a result, frequent packets have less memory access and, therefore, the total packet filtering time is reduced.

In 2013, *Zhong, Geng & Zhao (2013)* focused on a simple and very important form of remote authentication problem. In this form, membership requests for a dynamic set of n data elements which are stored in unknown directories are verified. In their study, some of the available methods for confirming membership requests such as the Merkle hash tree, skip list, and RSA tree were examined for the first time. In all of these methods, the data structures used by the algorithm to update the data are not fast enough and may have a high complexity time. It could also be possible to reconstruct a range of data structures during the update process. Therefore, they used the B+ tree data structure with RSA accumulators for the authentication scheme, which requires lower computational costs for membership queries in a dynamic data set.

*Trabelsi & Zeidan (2012)* provided in 2012 a mechanism to improve the filtering time of firewall packets by optimizing the comparison order of the matched security-rule fields to decide on the early rejection of incoming packets. Their proposed mechanism was based on changing the order of filtering fields according to traffic statistics. It also allowed to use multi-level classifying filters. Therefore, their proposed mechanism can be considered as a mechanism for protecting the device against denial-of-service attacks (DoS). Early packet acceptance is accomplished through the use of splay trees and changes dynamically with respect to traffic streams. Therefore, frequent packets have less memory access, thereby reducing the matching time. The purpose of their proposed method was to overcome some of the limitations of the previous technique called Self-adjusting Binary Search on Prefix Length (SA-BSPL). The numerical results of the simulation show that their proposed mechanism can improve the firewall performance in terms of total packet processing time compared to the SA-BSPL method.

*Zeidan & Trabelsi (2011)* in 2011 provided a mechanism to improve firewall performance through the rejection of denial-of-service attacks. To do this, they used a security policy of filtering as well as a statistical traffic plan that was implemented in the form of multi-level filter, splay tree, and hash tables. The proposed design rejects unwanted traffic and repetitive packets in the early stages and, therefore, less memory is used. As a result, packet matching time is generally reduced. The results of the evaluation of this method indicate that the proposed mechanism significantly reduces the processing time of DoS traffic.

*Trabelsi & Zeidan (2011)* explored firewall packet rejection techniques in 2011. Two of these techniques include FVSC and PBER that introduce the concept of approximate policy instead of using the full policy provided by the administrator. The benefit of such policies is that they are quicker at evaluating and adapting to dynamic traffic. The third technique, which is called SA-BSPL, uses the splay tree data structure. This data structure dynamically changes according to traffic behavior so that, when a node containing highly matching rules with packets is located close to the root, necessary actions on the packet are possible at a faster rate. These techniques allow the maximum number of packets to be processed as quickly as possible, thereby reducing the time of filtering process.

*Neji & Bouhoula (2008)* presented a dynamic packet routing algorithm in 2008. They considered a self-regulating tree by combining a binary search pattern on the prefix length with a splay tree. Using a set of hash tables and a splay tree, packet filtering was done according to the destination address. Their research paid particular attention to packets driven by the default path because it covered a major part of routers' traffic. Their design was better than previous models, especially for very diverse inputs, and had a logarithmic time cost for doing its tasks.

In 1995, *Kirschenhofer, Martínez & Prodinger (1995)* decided to mark the elements whose keys had been compared in the search algorithm in order to avoid unnecessary comparisons of the keys during the search in the skip list. Their evaluation criterion in this study was a detailed analysis of the total search cost (expectation and variance) so that the search cost would be calculated based on key-to-key comparisons and the results would be compared with standard search results. Their comparison shows that the cost of their method is much less than the standard search cost.

## Algorithms and tools

This section describes how the algorithms in question operate. Consider the sample rules in Table 1. This set of rules is arranged in descending order based on the fixed length of the source addresses, and if the source addresses are equal, the sorting operations are performed according to the destination addresses. Thus, the address placed at the top of the table has a higher priority than other addresses.

The set of the source and destination prefix addresses of the rules must be converted into a range of integers (*Trabelsi & Zeidan, 2012*). For this purpose, the upper and lower boundaries are first calculated for each prefix in the set of source addresses, as shown in Table 2. For the sake of simplicity, the prefix addresses are displayed in a six-bit format.

### Splay tree

For each field including the source address, destination address, source port number, destination port number, and protocol type, a splay tree should be created (*Trabelsi & Zeidan, 2011*; *Trabelsi & Zeidan, 2012*; *Trabelsi et al., 2015*). In addition to pointers to the left and right children as well as the parent node, each node of the tree contains a value and a counter to hold the number of times the node is matched with the input packets and a list for storing the rules. Initially, the counter of all nodes is set to zero. In the protocol tree, each node contains a list of rules whose protocol field has a value is equal to the value of the node, but in other trees each node contains a list of rules in which the lower boundary is less than or equal to the value of the node and the upper boundary is greater than or equal to the value of the node. As the values of the fields of source address, destination address, source port number, and destination port number have both upper and lower boundaries, they should be inserted into the corresponding trees in two steps (*Trabelsi & Zeidan, 2011*; *Trabelsi et al., 2015*).

In the first step, the lower boundary is inserted into the tree. If the lower boundary is less than the root value, it is inserted under the left tree, and if it is greater than the root value, it will be inserted under the right tree. Then the value of the lower boundary node

**Table 1  An example of a set of rules.** Each row of the table is a rule of a rule set. Each rule is represented as constraints on source IP, destination IP, source port number, destination port nummber and protocol.

| Rule | Source IP address | Destination IP address | Source port number | Destination port number | Protocol |
|------|------|------|------|------|------|
| R1 | 100* | 011* | 56,56 | 1024,65535 | 16 |
| R2 | 010* | 001* | 70,80 | 20,20 | 4 |
| R3 | 010* | 01* | 0,65535 | 2688,2688 | 16 |
| R4 | 000* | 10* | 56,80 | 0,65535 | 6 |
| R5 | * | * | 0,6553 | 2688,2688 | 16 |

**Table 2  An example of converting source prefix addresses to a numerical range.** Each prefix at the second column of the table is converted to corresponding upper and lower boundaries which are presented at the third and fourth columns. The fifth and sixth columns corresponds to decimal representation of the start and end points of the boundary.

| Rule | Source prefix addresses | Lower boundary | Upper boundary | Start | End |
|------|------|------|------|------|------|
| R1 | 100* | 100000 | 100111 | 32 | 39 |
| R2 | 010* | 010000 | 010111 | 16 | 23 |
| R3 | 010* | 010000 | 010111 | 16 | 23 |
| R4 | 000* | 100000 | 101111 | 32 | 47 |
| R5 | * | 000000 | 111111 | 0 | 63 |

is compared with the upper and lower boundary values of all the rules. When the lower boundary value lies within the range of a rule, the ID of that rule is added to the list of lower boundary rules. After being added to the tree, the lower boundary node will be moved to the root of the tree using the rotation operation. The second step is to insert the upper boundary into the tree. This step resembles the insertion of the lower boundary.

Figure 1 shows the steps for creating a splay tree for the source address fields of the rules in Table 2. In Fig. 1A, the R1 rule has been added to the tree. To this end, first the lower boundary value is inserted. Since the value of 32 lies within the range of R1 and R5, the ID of these rules is added to the rules list. Then, the value of 39 is inserted and the IDs of R1 and R5 rules are added to its rules list. Finally, 39 is transferred to the root of the tree with a left rotation. In Fig. 1B, the R2 rule has been added to the tree. In this case, the value of 16 is inserted. First the node 16 is searched in the tree and, if it is not found, it will be inserted in the correct place and the IDs of R2, R3, and R5 are added to the its rules list. Finally, the node 16 is transferred to the root through a right rotation between 23 and 39 and a right rotation between 16 and 32. In Fig. 1C, the value of 23, which is the lower boundary of R2, is inserted. In the next step, the R2, R3, and R5 rules are added to its list of rules. Then the node 23 is transferred to the root with through a right rotation between the nodes 23 and 32 and a left rotation between 23 and 16. Finally, the R3 rule is added to the tree. Since its values have already been added, no change occurs in the tree.

## Skip list

To build skip lists (*Pan et al., 2016*), the set of rules is first transmitted to the program and the upper and lower boundaries of the rules are calculated. For each of the fields of

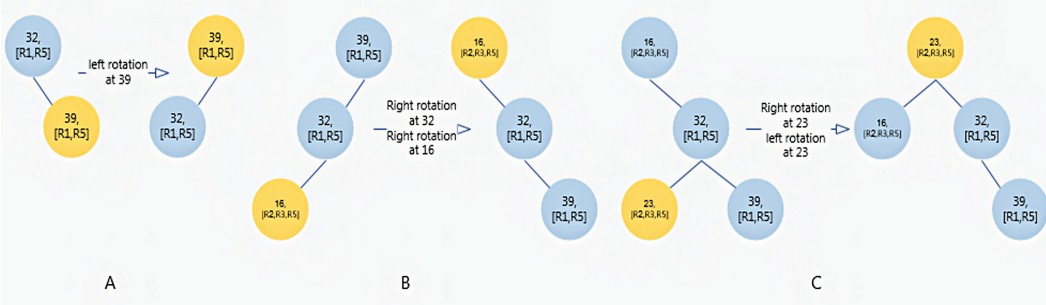

**Figure 1  The steps of creating a splay tree.** (A) Inserting the R1 rule; (B) inserting the lower boundary of the R2 rule; (C) inserting the upper boundary of the R2 rule.

source address, destination address, source port number, destination port number, and protocol type, a skip list must be created. Each skip list contains a value, a list for storing rules, and a list for storing pointers to subsequent nodes based on the level of each node. To determine the level of each node, a random function is used which creates an integer in a specified range (between 0 and 15 in our implementation). In the protocol skip list, each node contains a list of rules whose protocol field has a value equal to the value of the node, but in other skip lists each node contains a list of rules in which the lower boundary of the corresponding field is less than or equal to the value of the node and the upper boundary of the corresponding field is greater than or equal to the value of the node. As the values of the fields of source address, destination address, source port number, and destination port number have both upper and lower boundaries, they should be inserted into the corresponding skip lists in two steps. In the first step, the lower boundary is inserted into the skip list. Then the value of the lower boundary node is compared with the upper and lower boundary values of all the rules.

When the lower boundary value lies within the range of a rule, the ID of that rule is added to the list of lower boundary rules. Then, based on the node's level, a list of pointers is built for the created node. In the second step, we add the upper boundary. This step resembles the insertion of the lower boundary. Figure 2 shows the steps for creating a skip list for the source address field in Table 2. In Fig. 2, the R1 rule has been added to the skip list. First, the lower boundary of 32 at the level 0 is inserted into the skip list. Since the value of 32 lies within the range of R1 and R5, the ID of these rules is added to the rules list. Then the upper boundary of 39 at the level 2 is inserted and the IDs of R1 and R5 rules are added to its rules list. In Fig. 2B, the R2 rule has been added to the skip list. The value of 16 at the level 3 is inserted and the IDs of R2, R3, and R5 are added to its rules list. Next, the value of 23 at the level 1 is inserted and the IDs of R2, R3, and R5 are added to its rules list. In Fig. 2C, the R3 rule is added to the skip list. The values of this rule are repetitive.

## Packet classification

With both skip lists and splay trees, packet classification is as following. When a packet is received, the information of its header including source and destination address, source and destination port number, and protocol type are extracted. Next, for each of the mentioned

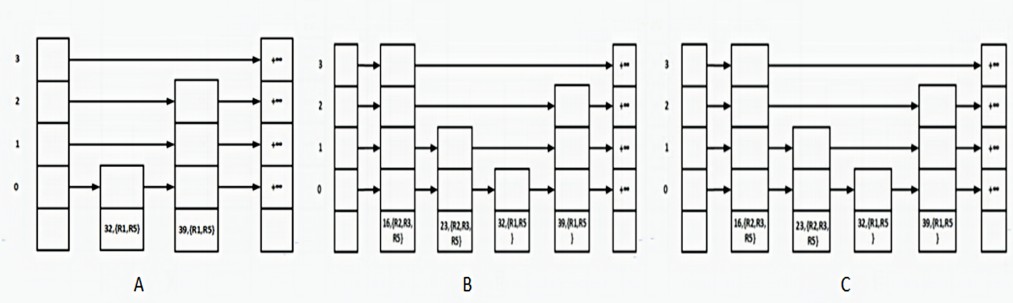

**Figure 2** **Steps to create skip list.** (A) Inserting the R1 rule; (B) inserting the R2 rule; (C) inserting the R3 rule.

fields in the packet, a skip list or a splay tree is created and searched simultaneously to find a matching node. Search results on any list or tree include a list of matching rules. In order to find a common rule between the five lists obtained from the splay trees, an intersection operation is performed between them. The result of the intersection may be null or contain several rules. If the result is null, the action associated with the default rule is applied to the packet; otherwise, the action related to the rule with the highest priority is applied. Because the rules were initially arranged according to priority, the rule with the smallest row number has the highest priority.

As mentioned in the previous section, splay trees are binary search trees that self-adjust so that the deepest met surviving node in any operation becomes the root following the operation. The splay tree stores no balance or weight information, but it performs many tree rotations after every access, which makes it less practically efficient than skip lists in many applications. These rotations can be particularly harmful when nodes are augmented with auxiliary structures. This situation is present in packet classification. Simple operations like move-to-root could partially solve this problem and improve the performance of the splay trees when there is locality of references in the operation sequence. But, it is not ideal in the case of packet classification where the sequence of the burst operations has no predictable locality (*Sahni & Kim, 2002*).

On the contrary, the simplicity of skip list algorithms makes them easier to implement and provides significant constant factor speed improvements over balanced tree and self-adjusting tree algorithms like splay trees (*Dean & Jones, 2007*). Their scheme is designed to give good expected performance for busty access patterns (*Sahni & Kim, 2002*). Skip lists are also very space efficient (*Sen, 1991*; *Kirschenhofer, Martínez & Prodinger, 1995*).

To practically investigate the above predictions about the performance of these two competitor algorithms, we implement and experiment them on several data sets.

## Implementation and evaluation

Splay tree and skip list approaches were implemented in C++ and executed ten times on a system with Intel Core i5 2.30 GHz and 4GB of RAM. The performance criteria were calculated using average results.

The two approaches were evaluated based on the number of memory accesses for packet classification, classification time, and memory usage. The Class Bench tool (*Taylor & Turner, 2007*) was used to generate rule sets and packet headers. The ACL and IPC rules were created in the evaluations to compare the number of memory accesses for packet classification as well as the times of packet classification with 500, 1 k, and 8 k rules. For our evaluations, we generated a set of 8 k, 32 k, and 128 k packet headers corresponding to each of the set of rules. We also used 1000 IPC and ACL rules to determine the amount of memory usage.

First, we look at packet classification time which is the time span from when a packet enters the structure of a classifier until the system can find the matching rule for that packet. The shorter the packet classification time, the more efficient the structure of the classifier will be. Figure 3 shows the time for classifying a wide variety of packets based on the sets of 500, 1k, and 8k ACL and IPC rules for the skip list and splay tree. Figure 3A compares these two approaches for 8 k packets. In these charts, the smallest difference between the two approaches is observed for 500 rules and the largest difference for 8k rules. The skip list classifies packets for 500 IPC and ACL rules in 391 and 1,415 ms, respectively, and the splay tree does this in 1,011 and 2,271 ms, respectively. Also, the packet classification time of the skip list for 8 k IPC and ACL rules is 805 and 4,231 ms, respectively, while this time for the splay tree is 684 and 8,131 ms, respectively. It can be concluded that, with an increased number of rules, the difference in classification time between the performances of the two approaches becomes greater. In fact, the skip list performs this task more optimally than the splay tree. Also, the type of rules plays an important role in packet classification time so that packets are classified in a significantly shorter time when matched with IPC rules. A decreased number of rules would reduce the time difference while increased number of rules would increase this difference. Consequently, the choice of the type of rules for packet classification might affect performance.

Figure 3B evaluates both the skip list and splay tree for 32k packets. As mentioned in Fig. 3A, the least difference in packet classification time between ACL and IPC rules is observed for 500 rules where as the largest difference is observed for 8 k rules. The skip list classifies packets for 500 IPC and ACL rules in 1,849 and 4,660 ms, respectively, and the splay tree does this in 5,981 and 7,994 ms, respectively. Also, the packet classification time of the skip list for 8 k IPC and ACL rules is 3,192 and 3,813 ms, respectively, while this time for the splay tree is 11,947 and 25,722 ms, respectively. As a result, with the increase in the number of packets, the skip list still outperforms splay tree in terms of packet classification time. However, increased number of packets has difference in packet classification time of the two approaches for 500 ACL rules smaller than that for 500 IPC rules. This means that, if the number of rules is small enough, an increased number of packets could be best handled by ACL rules; otherwise, IPC rules should be used for larger numbers of rules. The difference is particularly significant in classification with 8 k rules. In Fig. 3C, the results of the classification of 128 k packages are evaluated. In this evaluation, too, the skip list has a better performance than the splay tree. For the set of IPC rules, the smallest difference between the two approaches can be observed for 1 k rules. In this case, the skip list classifies packets in 6,792 ms and the splay tree does this in 11,031 ms. As in the previous part, the

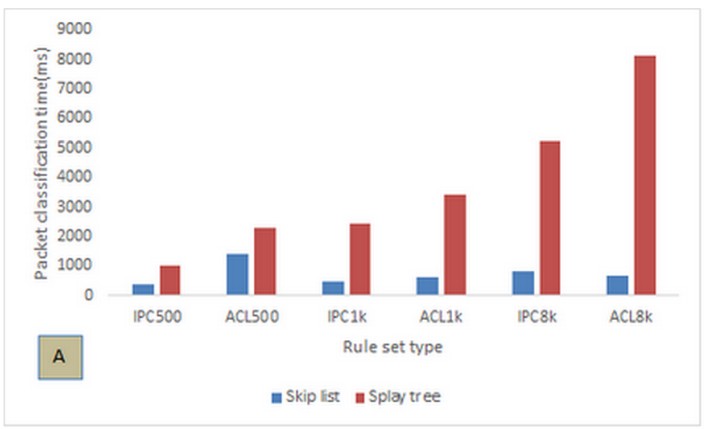

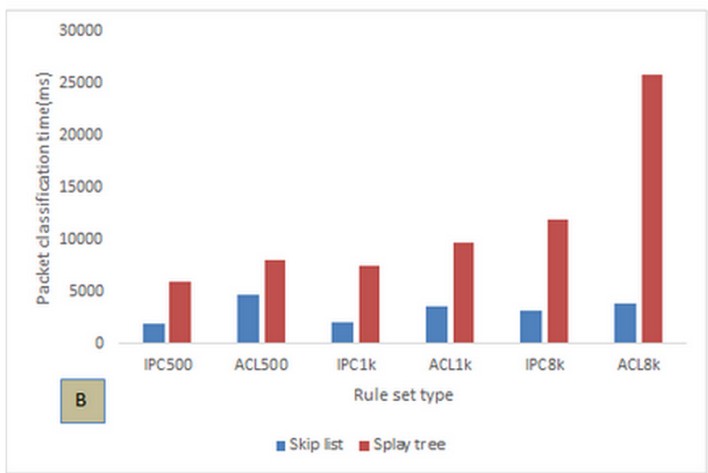

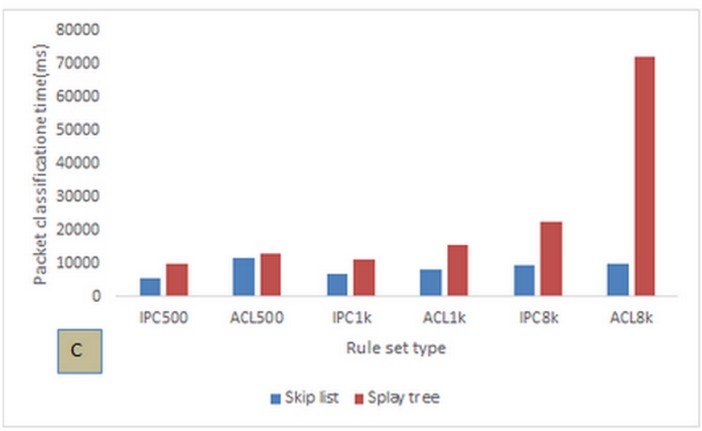

**Figure 3** **Packet classification time for the sets of 500, 1k, and 8k ACL and IPC rules for different numbers of packets.** (A) 8k, (B) 32k, and (C) 128k packets.

smallest difference between these two approaches is observed for the set of 500 ACL rules so that the packet classification time of the skip list is 11,846 ms whereas that of the splay tree is 13,020 ms. The difference in packet classification time between skip list and splay tree with both IPC and ACL rules is significant. This result can be used to select appropriate rules for designing a system that is to be efficient in terms of packet classification. Here again, the greatest difference between the two approaches is manifested in the case of 8 k rules. With 8 k IPC and ACL rules, the skip list classifies packets in 9,508 and 9,778 ms, respective, while splay tree does this task in 22,457 and 72,444 ms, respectively. As can be seen, this time difference for the IPC rules is much smaller than for the ACL rules. In general, Fig. 3 shows that skip list approach classifies packets in a shorter time. In addition, the increase in the packet classification time of the skip list due to increased number of rules is significantly less than that of the splay tree. It can be inferred that skip list has a better performance than the splay tree in the classification of packets.

One of the most important criteria for the performance of classification approaches is the speed of search. In the architecture of network processors, memory access is the most important reason for prolonged execution of commands on packets. Frequent access to memory reduces system performance. Reduced memory access would decrease packet classification time and, thus, accelerate the process. Therefore, decreased memory access is central to the efficiency of an approach.

Figure 3A evaluates the two approaches for 8k packets. As can be seen, in all cases skip list has fewer memory accesses than splay tree. Also, the minimum number of memory access is 65,477, which belongs to skip list with 500 IPC rules. Splay tree has 477,664 memory accesses with 8 k ACL rules, which is the highest number of access in our evaluation. With the increase in the number of rules, the difference in memory access between the two approaches increases significantly. With 8 k IPC and ACL rules, the skip list has 104,017 and 98,476 memory accesses, respectively, while the splay tree accesses memory 198,664 and 477,664 times, respectively. The greatest difference in the number of memory accesses between the skip list and splay tree is observed in the case of 8 k ACL rules in which splay tree accesses memory 379,188 times more than skip list. Figure 3B compares skip list and splay tree for 128 k packets. As in previous parts, the skip list outperforms the splay tree in terms of memory access. In general, the minimum number of memory access is 215,169 which belongs to the skip list with 500 IPC rules. The maximum number is 1890056 which belongs to the splay tree with 8 k ACL rules. The greatest difference in the number of memory accesses between the skip list and the splay tree is observed in the case of 8 k ACL rules in which the splay tree accesses memory 1496966 times more than the skip list. It can be observed in the chart that the number of memory accesses for both approaches using IPC rules is much smaller than using ACL rules, which could be a reason for preferring IPC rules in the design of such systems. Figure 3C compares the skip list and the splay tree with 128 k packets. The chart shows that, with increase in the number of packets with different numbers of rules, the skip list has less memory access than does the splay tree. In this chart, the smallest number of memory access is 1061018 which belongs to the skip list with 500 IPC rules and the largest number of access is 8024198 which belongs to the splay tree with 8 k ACL rules. The greatest difference between the two approaches is observed in

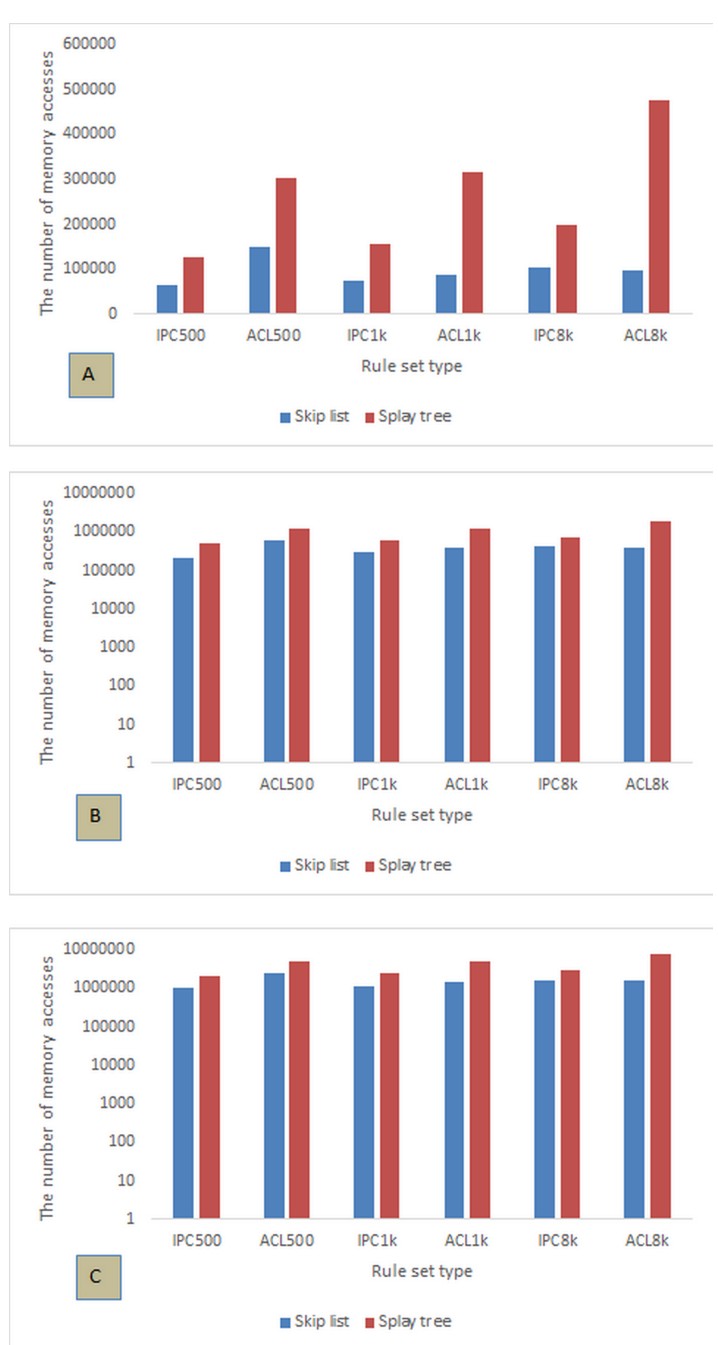

**Figure 4** **The number of memory accesses for packet classification with sets of 500, 1,000, and 8,000 ACL and IPC rules for different number of packets.** (A) 8k, (B) 32k, and (C) 128k packets.

the case of 8 k ACL rules, with the splay tree having accessed memory 6356259 times more than the skip list. As can be seen in Fig. 4, the skip list has a better performance than the splay tree in terms of memory access. Also, with increasing number of rules, the increase in the number of memory accesses for the skip list is much smaller than that of the splay

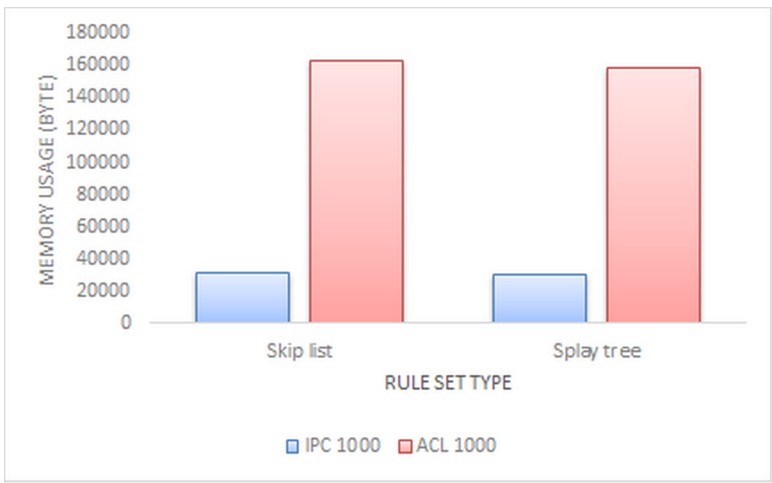

**Figure 5** **Memory usage for 1k ACL and IPC rules.** The red and blue bars represent the memory usage of the splay tree and skip list algorithms respectively.

tree. As a result, the performance of the skip list can be considered as more efficient than the splay tree. According to the results of the charts in Figs. 3 and 4, another important point is correspondence in the results of memory access time and number which exactly confirm each other in all cases.

Given the memory limitations in the majority of systems and the high costs of upgrading memories, another performance criterion for classification approaches is the amount of memory usage. As a result, every approach should aim at reducing memory usage. Figure 5 shows the amount of memory used in bytes by skip list and splay tree for classifying packets with 1 k ACL and IPC rules. As can be seen, the amount of memory used by skip list is 31,700 bytes with IPC rules and 162,960 bytes for ACL rules whereas the memory usage of splay tree is 30,528 bytes for IPC rules and 158,440 bytes for ACL rules. The amount of memory used by skip list with both sets of rules is slightly more than splay tree.

This additional amount of space is used to hold pointers in a skip list. Also, the amount of memory used by both approaches with IPC rules is significantly less than the memory used with ACL rules. However, this additional space can be reasonably justified by significant reduction in the number of memory accesses and packet classification time in skip lists.

## CONCLUSION

Packet classification is among the basic processes in network processors. The most important issue is the use of a packet classification approach that can keep up with the network speed. Such an approach should also optimize memory consumption. The existing methods have not been able to balance the time and memory consumption. On the other hand, binary trees work well when the elements enter accidentally, but they become inefficient in cases where the operations are sequential. In this study, therefore, we focused on the skip list and the splay tree and evaluated these two approaches with ACL and IPC rules. Our results suggest that skip list performs better in terms of package

classification time and the number of memory accesses. Also, with increase in the number of rules, packet classification time and memory access increase less in a skip list than in a splay tree. The amount of memory used by the skip list is slightly more than the splay tree, which is due to storing the pointers in skip lists. However, this additional space can be reasonably justified by significant reduction in the number of memory accesses and packet classification time in skip lists. Accordingly, the skip list can be considered as superior to the splay tree. Obviously, the data and control dependencies in the algorithms will change their performance in parallel processing. Therefore, the authors aim to study the parallelization of both algorithms on graphics processors and evaluate the performance of their parallel versions in further research.

### Funding
The authors received no funding for this work.

### Competing Interests
The authors declare there are no competing interests.

### Author Contributions
- Navid Khezrian performed the experiments, analyzed the data, contributed reagents/materials/analysis tools, prepared figures and/or tables, performed the computation work, approved the final draft.
- Mahdi Abbasi conceived and designed the experiments, analyzed the data, contributed reagents/materials/analysis tools, performed the computation work, authored or reviewed drafts of the paper, approved the final draft, submission of the paper.

### Data Availability
The raw measurements are available in Supplemental Files.

### Supplemental Information
Supplemental information for this article can be found online at http://dx.doi.org/10.7717/peerj-cs.204#supplemental-information.

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
