# Peer review of "Comparison of the performance of skip lists and splay trees in classification of internet packets"

_PeerJ Computer Science, doi:10.7717/peerj-cs.204_

## Round 0.1 · original submission · Minor Revisions

Reviewers have now commented on your paper. You will see that they are advising that you revise your manuscript. For your guidance, reviewers' comments are appended below.

Reviewer 1 ·

Basic reporting

1. Some typos, e.g., Line 59 freat--> great. A spell check should be applied to the document.

2. The paper talks about comparing Splay trees to skip lists for packet classification. The original paper by Sleator and Tarjan [1985, Self-adjusting binary search trees] that invented Splay trees should be cited.

Experimental design

1. It seems like the Splay tree is always worse than Skip List. Are there any scenarios, perhaps artificially constructed, when they may be more suitable?

2. If Skip lists are always better, perhaps a theoretical analysis will prove this.

3. What if the authors used a static binary tree (no rotations)? This will demonstrate whether the problem is with rotation rule of the Splay trees, or with using a tree data structure in the first place.

4. Generally, skip lists act as the same as binary trees, so in case the Splay rotation rule is what causes the inferior performance, what about simpler rules, like move to root?

Validity of the findings

No comment.

Reviewer 2 ·

Basic reporting

The authors present a comparative analysis of two decision tree implementations — splay trees and skip lists in order to determine which of these two are better suited for classification of internet packages. They find that skip lists perform better than splay trees in terms of package classification speed and memory accesses. They, however, note that the memory used by skip lists is higher than splay trees. Although based on Figure 5, it seems that they are not drastically different.

The paper is decently written, and is easy to follow. I would perhaps structure the tables and figures in a different way; at this point, they are at the end of the paper and completely detached from the experimental results section, which means I had to go back and forth, making it harder to read. Also, I do not see any reason why the table labels and the table itself have to be on different pages, given that both pages are more than half empty. Although, I am no expert in this area, the literature review does seem to be thorough. I find the citations to be a little strange, where the author names appear and again within parenthesis the last names with the year are given; I would think that providing just the year within the parenthesis ought to be enough. But these are all a matter in exposition and I would not dwell on them further.

Experimental design

My main concern is that there is no bench-marking with respect to the existing techniques for package classification. I believe that this being an important problem, there must be other well established techniques. Therefore I feel such a comparison analysis is almost imperative.

Validity of the findings

The other issue is the lack of novelty in the paper. It simply compares two decision tree implementations based on memory footprint and classification time.

In most practical scenarios skip lists often outperform most balanced search trees, and that “worst case” scenarios are typically infrequent in probabilistic data structures (such as skip list). Hence, I do not see why we need to use package classification to differentiate between the two. Is there some structure to package classification that I am missing? If there is, it would be beneficial if the authors could elucidate that clearly.

Additionally, splay lists attain O(log n) in the amortized sense. Have the authors considered this for the splay tree structure? Have they considered other form of (strictly) balanced search trees?

Lastly, I see that the authors have talked about parallelization of the algorithms. Skip lists are better for parallel implementation and writing multi-threaded programs (than most balanced search trees) as one has to only obtain the lock locally when it comes to modification or search. In most (balanced) search trees, locking is painful mainly due to re-balancing/splaying. Hence, I do not see why the outcomes over there would be any different (i.e., skip lists would still vastly outperform splay trees). Having said that it would have been a welcome addition to the papers as it would have boosted the contribution.

Additional comments

Overall, I feel that for the paper to be accepted, the authors must address the above concerns.

---

## Round 0.2 · accepted · Accept

Your revised manuscript is accepted for publication.